# InfoCD: A Contrastive Chamfer Distance Loss for Point Cloud Completion

**Fangzhou Lin**[1,2*]   **Yun Yue**[1*]   **Ziming Zhang**[1†]   **Songlin Hou**[1,3]   **Kazunori D Yamada**[2]
**Vijaya B Kolachalama**[4]   **Venkatesh Saligrama**[4]
[1]Worcester Polytechnic Institute, USA   [2]Tohoku University, Japan
[3]Dell Technologies, USA   [4]Boston University, USA
`{flin2, yyue, zzhang15, shou}@wpi.edu, yamada@tohoku.ac.jp,`
`{vkola, srv}@bu.edu`

## Abstract

A point cloud is a discrete set of data points sampled from a 3D geometric surface. Chamfer distance (CD) is a popular metric and training loss to measure the distances between point clouds, but also well known to be sensitive to outliers. We propose *InfoCD*, a novel contrastive Chamfer distance loss, and learn to spread the matched points to better align the distributions of point clouds. As such *InfoCD* leads to an improved surface similarity metric. We show that minimizing InfoCD is equivalent to maximizing a lower bound of the mutual information between the underlying geometric surfaces represented by the point clouds, leading to a *regularized* CD metric which is robust and computationally efficient for deep learning. We conduct comprehensive experiments for point cloud completion using InfoCD and observe significant improvements consistently over all the popular baseline networks trained with CD-based losses, leading to new state-of-the-art results on several benchmark datasets. Demo code is available at `https://github.com/Zhang-VISLab/NeurIPS2023-InfoCD`.

## 1   Introduction

**Point Cloud Completion.** Point clouds, one of the most important data representations that can be easily acquired, play a key role in modern robotics and automation applications [1–3]. However, raw data of point clouds captured by existing 3D sensors are usually incomplete and sparse due to occlusion, limited sensor resolution, and light reflection [4–8], which can negatively impact the performance of downstream tasks that require high-quality representation, such as point cloud segmentation and detection. *Point cloud completion* [9] refers to the task of inferring the complete shape of an object or scene from incomplete raw point clouds. Recently, many (deep) learning based approaches have been introduced to point cloud completion ranging from supervised learning, self-supervised learning to unsupervised learning [10–15]. Among these methods, supervised learning with a general encoder-decoder structure is the prevailing architectural choice for many researchers, consistently achieving state-of-the-art results on mainstream benchmarks [16, 17, 8, 1, 18].

**Learning with Chamfer Distance (CD).** CD is a commonly employed metric in point cloud completion, for example in studies like [19, 20]. It assesses the dissimilarity in shape between two sets of point clouds by calculating the average distance from each point in one set to its nearest neighbor in the other set. Minimizing the Euclidean distances between matched points, as done in CD, is a known method that is sensitive to outliers, resulting in *clumping behavior* that involves

---

*Co-first authors.

†Corresponding author.

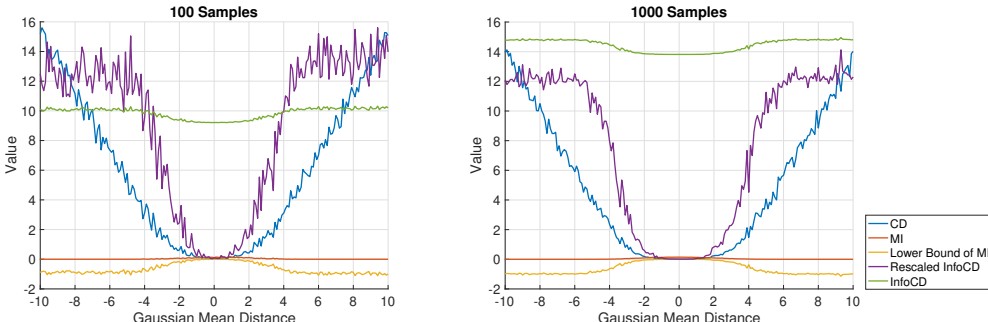

Figure 1: Illustration of comparison among CD, MI, and InfoCD with different numbers of samples.

a substantial number of many-to-one correspondences in point matching, forming small clusters visually. These observations can significantly deviate from the assumption of uniform sampling from the underlying geometric surfaces, which is commonly used to generate point clouds.

**Learning with Mutual Information (MI).** A more fundamental problem in measuring point cloud similarity (or distance alternatively) is: *How can we measure the similarities between the underlying geometric surfaces represented by the point clouds?* To address this issue, one potential method is to compute their MI by taking each point cloud as a discrete set of point samples from a random variable following a certain distribution, and then plugging both point clouds into the MI formula to measure the similarity of the two random variables. However, two limitations prevent us from using it directly as a loss function in learning: (1) It may be nontrivial to define the joint probability distribution between two point clouds since we do not have any prior knowledge, and (2) The requirement for extensive memory, necessary for handling a large volume of data points and deep learning models, could render the practical implementation of the MI (Mutual Information) loss unfeasible.

**Our Approach and Contributions.** Motivated by contrastive learning, we propose a novel contrastive Chamfer distance loss, namely *InfoCD*, to learn to spread the matched points for better distribution alignments between point clouds as well as accounting for surface similarity estimation. Similar to InfoNCE [21], our InfoCD minimization tends to maximize a lower bound of the MI between the underlying geometric surfaces represented by the point clouds. This indeed leads to a regularized CD loss that can be minimized effectively and efficiently with similar computational complexity to CD.

We further illustrate our high level ideas in Fig. 1. To plot such figures, we consider generating point clouds uniformly sampled from two 1D Gaussian distributions both with unit variance but different means, one fixed at 0 and the other varying from -10 to 10 at step-size of 0.1. We then compute CD, InfoCD (see Eq. 5 with $\tau = 1$), the lower bound of MI (see Eq. 8), and MI where the joint probability distribution is computed based on the exponential of negative Euclidean distances. To better view the difference between the curves of CD and InfoCD, we also rescale the InfoCD curve by aligning its minimum and maximum values with those for CD based on linear scaling so that their value ranges are the same. With different numbers of samples, we can clearly see that: (1) InfoCD is more robust to outliers (*i.e.,* the cases far away from 0 mean) than CD by penalizing larger deviations with some similar numbers (though smaller numbers of samples have more fluctuations); (2) The sharper purple curves around 0 indicate that InfoCD may lead to better convergence; (3) The lower bound based on InfoCD can approximate MI well with the maximum at 0, which indicates that InfoCD can be taken as a good MI estimator and lead to better point distributions with fewer visual clusters.

To summarize, we list our main contributions as follows:

- We propose InfoCD by introducing contrastive learning into the CD loss, leading to a regularized CD loss for better point distribution alignments.
- We analyze the connection between InfoCD and MI as well as learning behavior with InfoCD.
- We achieve state-of-the-art results on popular benchmark datasets for point cloud completion.

## 2   Related Work

**Point Cloud Completion.** As the first learning-based point cloud completion network, PCN [10] extracts global features in a similar way PointNet [22] did and generates points through folding

operations as in FoldingNet [23]. In order to obtain local structures among points, Zhang et al. [24] proposed extracting multi-scale features from different layers in the feature extraction part to enhance the performance. CDN [25] uses a cascaded refinement network to bridge the local details of partial input and the global shape information together. Lyu et al. [26] proposed treating point cloud completion as a conditional generation problem in the framework of denoising diffusion probabilistic models (DDPM) [27]. Attention mechanisms such as Transformer [28], demonstrate their superiority in capturing long-range interaction as compared to CNNs' constrained receptive fields. For instance, to preserve more detailed geometry information for point cloud generation in the decoder, SA-Net [29] uses the skip-attention mechanism to merge local region information from the encoder and point features of the decoder. SnowflakeNet [17] and PointTr [16] pay extra attention to the decoder part with Transformer-like designs. PointAttN [1] was proposed solely based on Transformers.

**Distance Metrics for Point Clouds.** Distance in point clouds is a non-negative function that measures the dissimilarity between them. Since point clouds are inherently unordered, the shape-level distance is typically derived from statistics of pair-wise point-level distances based on a particular assignment strategy [20]. With relatively low computational cost fair design, CD and its variants are extensively used in learning-based methods for point cloud completion tasks [30, 26, 31, 32]. Earth Mover's Distance (EMD), which is another widely used metric, relies on finding the optimal mapping function from one set to the other by solving an optimization problem. In some cases, it is considered to be more reliable than CD, but it suffers from high computational overhead and is only suitable for sets with exact numbers of points [33, 34]. Recently, Wu et al. [20] propose a Density-aware Chamfer Distance (DCD) as a new metric for point cloud completion which can balance the behavior of CD and computational cost in EMD to a certain level. Lin et al. [35] proposed a HyperCD that computes CD in a hyperbolic space and achieves significantly better performance than DCD.

**Contrastive Learning.** Recently, learning representations from unlabeled data in contrastive way [36, 37] has been one of the most competitive research fields [21, 38–50]. Popular model structures like SimCLR [42] and Moco [45] apply the commonly used loss function InfoNCE [21] to learn a latent representation that is beneficial to downstream tasks. Several theoretical studies show that contrastive loss optimizes data representations by aligning the same image's two views (positive pairs) while pushing different images (negative pairs) away on the hypersphere [51–54]. A good survey on contrastive learning can be found in [55]. More recently, contrastive learning has found its way into point cloud applications as well. For instance, Tang et al. [56] proposed a contrastive boundary learning framework for point cloud segmentation. Yang et al. [57] proposed the mutual attention module and co-contrastive learning for point cloud object co-segmentation. Jiang et al. [58] proposed a guided point contrastive loss to enhance the feature representation and model generalization ability in semi-supervised settings for point cloud segmentation. Du et al. [59] proposed a self-contrastive learning approach for self-supervised point cloud representation learning. Wang et al. [60] proposed exploring whether maximizing the mutual information across shallow and deep layers is beneficial to improve representation learning on point clouds, leading to a new design of Maximizing Mutual Information (MMI) Module. Afham et al. [61] proposed CrossPoint, a simple cross-modal contrastive learning approach to learn transferable 3D point cloud representations with 2D images. Shao et al. [62] proposed a spatial consistency guided network (SCRnet) using contrastive learning for point cloud registration.

## 3 InfoCD Loss

### 3.1 Preliminaries

**InfoNCE Loss.** In [21], the InfoNCE loss is defined as follows:

$$\mathcal{L}_{\text{InfoNCE}} = -\sum_{x} \log \left\{ \frac{\exp\left\{\frac{1}{\tau} s(x^+, x; \theta)\right\}}{\exp\left\{\frac{1}{\tau} s(x^+, x; \theta)\right\} + \sum_{x^-} \exp\left\{\frac{1}{\tau} s(x^-, x; \theta)\right\}} \right\}, \quad (1)$$

where $x, x^+, x^-$ denote the anchor, its positive and negative samples, $s$ denotes a similarity function parametrized by $\theta$, and $\tau \geq 0$ is a predefined temperature that controls the sharpness.

**Proposition 1** (InfoNCE *vs.* MI [21]). *Let $c_t$ be the context at the $t$-th time step, and $x_{t+k}$ be the future target. Then given a set of $N$ random samples, $\{x_1, \cdots, x_N\}$, containing one positive sample from the distribution $p(x_{t+k}|c_t)$ and $N-1$ negative samples from the "proposal" distribution*

$p(x_{t+k})$, *we have*

$$I(x_{t+k}, c_t) \geq \log(N) - \mathcal{L}_{InfoNCE}, \tag{2}$$

*where I denotes the mutual information (MI) operator.*

Please refer to the appendix in [21] for the proof. This proposition provides us with an alternative way to measure MI approximately and implicitly. If point clouds can fit into the setting of InfoNCE, we then may better estimate the underlying surface similarities from point clouds.

**Chamfer Distance Loss.** In the sequel, we denote $(x_i, y_i)$ as the $i$-th point cloud pair, with $x_i = \{x_{ij}\}$ and $y_i = \{y_{ik}\}$ as two sets of 3D points, and $d(\cdot, \cdot)$ as a certain distance metric. Then the CD loss for point clouds can be defined as follows:

$$\mathcal{L}_{\text{CD}}(x_i, y_i) = \ell_{\text{CD}}(x_i, y_i) + \ell_{\text{CD}}(y_i, x_i) = \frac{1}{|y_i|} \sum_k \min_j d(x_{ij}, y_{ik}) + \frac{1}{|x_i|} \sum_j \min_k d(x_{ij}, y_{ik}), \tag{3}$$

where $|\cdot|$ denotes the cardinality of a set. For point cloud completion, function $d$ usually refers to

$$d(x_{ij}, y_{ik}) = \begin{cases} \|x_{ij} - y_{ik}\| & \text{as } \textit{L1-distance} \\ \|x_{ij} - y_{ik}\|^2 & \text{as } \textit{L2-distance} \end{cases} \tag{4}$$

where $\|\cdot\|$ denotes the Euclidean $\ell_2$ norm of a vector.

### 3.2 Our Loss Function

Given the considerations above, we propose the following formula as our InfoCD loss:

$$\mathcal{L}_{\text{InfoCD}}(x_i, y_i) = \ell_{\text{InfoCD}}(x_i, y_i) + \ell_{\text{InfoCD}}(y_i, x_i), \text{ where}$$

$$\ell_{\text{InfoCD}}(x_i, y_i) = -\frac{1}{|y_i|} \sum_k \log \left\{ \frac{\exp\{-\frac{1}{\tau'} \min_j d(x_{ij}, y_{ik})\}}{\sum_k \exp\{-\frac{1}{\tau} \min_j d(x_{ij}, y_{ik})\}} \right\}$$

$$\propto \frac{1}{\tau |y_i|} \sum_k \min_j d(x_{ij}, y_{ik}) + \lambda \log \left\{ \sum_k \exp \left\{ -\frac{1}{\tau} \min_j d(x_{ij}, y_{ik}) \right\} \right\} \tag{5}$$

$$= -\frac{1}{|y_i|} \sum_k \log \left\{ \frac{\exp\{-\frac{1}{\tau} \min_j d(x_{ij}, y_{ik})\}}{\left[ \sum_k \exp\{-\frac{1}{\tau} \min_j d(x_{ij}, y_{ik})\} \right]^\lambda} \right\} \tag{6}$$

$$\implies \mathcal{L}_{\text{InfoCD}}(x_i, y_i) \propto \frac{1}{\tau} \mathcal{L}_{\text{CD}}(x_i, y_i) + \lambda \mathcal{R}(x_i, y_i) \tag{7}$$

with $\mathcal{R}(x_i, y_i) = \log \left\{ \sum_{m,n} \exp \left\{ -\frac{1}{\tau} [\min_j d(x_{ij}, y_{in}) + \min_k d(x_{im}, y_{ik})] \right\} \right\}$ as a regularizer and $\lambda = \frac{\tau'}{\tau} \in (0, 1]$ as a predefined constant controlling the trade-off between the loss and the regularizer. The smaller $\mathcal{L}_{\text{CD}}(x_i, y_i)$ is, the larger $\mathcal{R}(x_i, y_i)$ is accordingly. From this perspective, we can easily see that our InfoCD loss is equivalent to a regularized CD loss.

### 3.3 Analysis

**InfoCD *vs.* MI.** To see the connections between InfoCD and MI, we have the following lemma:

**Lemma 1.** *Consider two point clouds $x_i = \{x_{ij}\}, y_i = \{y_{ik}\}$ representing two underlying geometric surfaces $\mathcal{X}_i, \mathcal{Y}_i$. Now we introduce a new random variable $z_{y_{ik}}$ whose probability distribution $p(z_{y_{ik}} | \mathcal{X}_i = \mathcal{Y}_i)$ indicates how likely a point $y_{ik}$ can be sampled from $\mathcal{Y}_i$ conditional on $\mathcal{X}_i = \mathcal{Y}_i$, and "proposal" distribution $p(z_{y_{ik}})$ indicates the likelihood of generating an arbitrary $y_{ik}$. With these notations, we will have*

$$I(z_{y_{ik}}; \mathcal{X}_i = \mathcal{Y}_i) \geq \log(|y_i|) - \ell_{InfoCD}(x_i, y_i). \tag{8}$$

*Proof.* By following Prop. 1 and the proof in [21], we can take $z_{y_{ik}}, \mathcal{X}_i = \mathcal{Y}_i$ as the replacements for $x_{t+k}, c_t$. Then samples from $p(z_{y_{ik}} | \mathcal{X}_i = \mathcal{Y}_i)$ will be "positive", and ones from $p(z_{y_{ik}})$ will be

"negative" in the language of contrastive learning. Now we can construct $|y_i|$ groups, as illustrated in Fig. 2, where each group consists of the entire point cloud $x_i$ and a point in $y_i$. We further can take an arbitrary group as a positive and the rest as negatives. Finally, by parametrizing $\frac{p(z_{y_{ik}}|\mathcal{X}_i=\mathcal{Y}_i)}{p(z_{y_{ik}})} \propto \exp\left\{-\frac{1}{\tau'}\min_j d(x_{ij}, y_{ik})\right\}$ for positives and similarly to negatives, we can complete our proof. $\square$

Therefore, based on this lemma, minimizing InfoCD tends to better estimate the lower bound of MI that indicates the underlying surface similarities between point clouds.

**Point Spread in Learning & Testing.** To better understand the behavior of different losses, we first introduce another lemma as follows:

**Lemma 2.** *Consider an optimization problem $\min_{\omega\in\Omega}\sum_i g(h(x_i;\omega))$ where $h:\mathcal{X}\times\Omega\to\mathbb{R}, g:\mathbb{R}\to\mathbb{R}$ are both Lipschitz continuous functions, $h$ is also smooth over $\omega$, and $x_i\in\mathcal{X}$ is a data point. Then based on gradient descent, i.e., $\omega_{t+1}=\omega_t-\eta_t\sum_i\nabla g(h(x_i;\omega_t))$ with a learning rate $\eta_t\geq 0$ at the t-th iteration and a gradient operator $\nabla$, it holds that given a new data point $\tilde{x}$,*

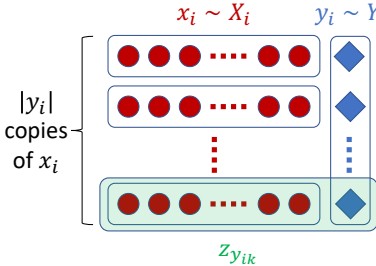

Figure 2: Illustration of group construction process for MI estimation.

$$h(\tilde{x};\omega_t) - h(\tilde{x};\omega_{t+1}) \approx \sum_i \eta_t \left.\frac{\partial g}{\partial h}\right|_{(x_i;\omega_t)} \nabla h(x_i;\omega_t)^T \nabla h(\tilde{x};\omega_t) = \eta_t \sum_i \left.\frac{\partial g}{\partial h}\right|_{(x_i;\omega_t)} \kappa(x_i,\tilde{x};\omega_t),$$
(9)

*where $\omega_0$ is the initialization of $\omega$, $\kappa$ denotes a (neural) tangent kernel function parametrized by $\omega_t$, $(\cdot)^T$ is the matrix transpose operator, and $\left.\frac{\partial g}{\partial h}\right|_{(x_i;\omega_t)}$ is the derivative of $g$ over $h$ at point $(x_i;\omega_t)$.*

*Proof.* Using the linear approximation of $h$ and the assumptions, we can easily prove this lemma. $\square$

To connect this lemma with CD and InfoCD, let us first compute the gradients of $\ell_{\text{CD}}$ and $\ell_{\text{InfoCD}}$:

$$\nabla\ell_{\text{CD}}(x_i,y_i) = \sum_k \frac{1}{|y_i|}\nabla d(x_{ik'},y_{ik}),$$
(10)

$$\nabla\ell_{\text{InfoCD}}(x_i,y_i) = \frac{1}{\tau}\sum_k\left[\frac{1}{|y_i|}-\tau'\frac{\exp\{-\frac{1}{\tau}d(x_{ik'},y_{ik})\}}{\sum_k\exp\{-\frac{1}{\tau}d(x_{ik'},y_{ik})\}}\right]\nabla d(x_{ik'},y_{ik}),$$
(11)

where $k'=\arg\min_j d(x_{ij},y_{ik}),\forall k$. By viewing $d$ as $h$ and $\ell_{\text{CD}}$ (or $\ell_{\text{InfoCD}}$) as $g$, we can see the weight $\frac{\partial\ell_{CD}(x_{ik'},y_{ik};\omega_t)}{\partial d}=\frac{1}{|y_i|}>0$ but $\frac{\partial\ell_{InfoCD}(x_{ik'},y_{ik};\omega_t)}{\partial d}=\frac{1}{|y_i|}-\tau'\frac{\exp\{-\frac{1}{\tau}d(x_{ik'},y_{ik})\}}{\sum_k\exp\{-\frac{1}{\tau}d(x_{ik'},y_{ik})\}}\in\mathbb{R}$. To simplify the analysis and explanation, assuming that the normalized gradients, $\frac{\nabla d(\cdot,\cdot)}{\|\nabla d(\cdot,\cdot)\|}$, can be viewed as random samples from a high dimensional (*i.e.,* the number of network parameters that is much larger than the number of samples in mini-batches) normal distribution, then for two different random inputs $(x_{im'},y_{im}),(x_{in'},y_{in})$, it will be expected [63] that the corresponding gradients will be close to being orthogonal to each other, *i.e.,* $\nabla d(x_{im},y_{ik})^T\nabla d(x_{in},y_{ik})\approx 0$. Now by substituting this assumption into Eq. 9, we can have

$$\ell_{\text{CD}}(x_i,y_i;\omega_t) - \ell_{\text{CD}}(x_i,y_i;\omega_{t+1}) \approx \eta_t\sum_k\frac{\partial\ell_{\text{CD}}(x_{ik'},y_{ik};\omega_t)}{\partial d}\|\nabla d(x_{ik'},y_{ik})\|^2,$$
(12)

$$\ell_{\text{InfoCD}}(x_i,y_i;\omega_t) - \ell_{\text{InfoCD}}(x_i,y_i;\omega_{t+1}) \approx \frac{\eta_t}{\tau}\sum_k\frac{\partial\ell_{\text{InfoCD}}(x_{ik'},y_{ik};\omega_t)}{\partial d}\|\nabla d(x_{ik'},y_{ik})\|^2,$$
(13)

with another assumption that the matched point pairs keep unchanged over iterations. From this perspective, we can easily see that the *negative* weights of $\frac{\partial\ell_{InfoCD}(x_{ik'},y_{ik};\omega_t)}{\partial d}$ with smaller distances will push the predicted points away from the matched ground-truth points, while *positive* weights will tend to reduce the distances in both metrics, as illustrated in Fig. 3. In this contrastive way, the predicted points will be more likely to be spread for better alignment with the ground-truth points.

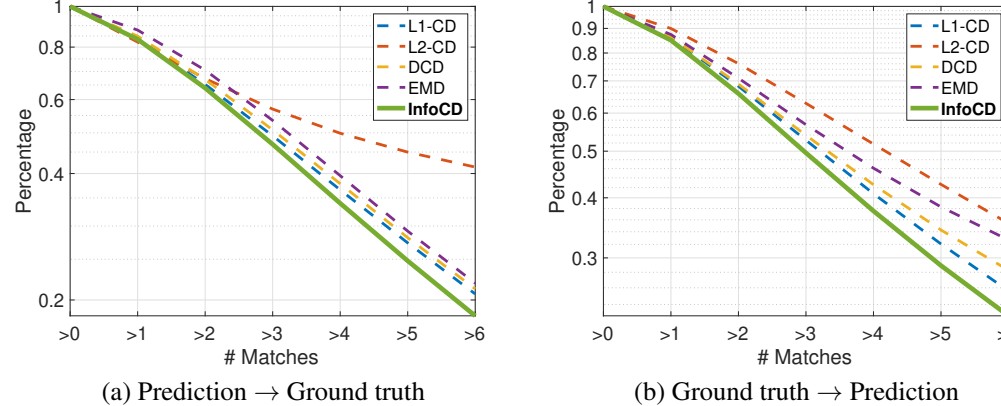

(a) Prediction → Ground truth        (b) Ground truth → Prediction

Figure 4: Comparison on point percentage *vs.* the number of matches per point using CP-Net [64] on ShapeNet-Part [65] dataset, where the CP-Net is trained with five different loss functions. Clearly, InfoCD has better point spread and thus distribution alignments.

To demonstrate the capability of our InfoCD loss to spread the matched points at test time, we illustrate some comparison results in Fig. 4, where we can see clearly that InfoCD can significantly reduce the numbers of many-to-one matched points, leading to better point distribution alignments.

**Convergence.** In general, there is no guarantee that training networks with InfoCD using (stochastic) gradient descent will converge if the matched point pairs between the predictions and ground truth are frequently changed. However,

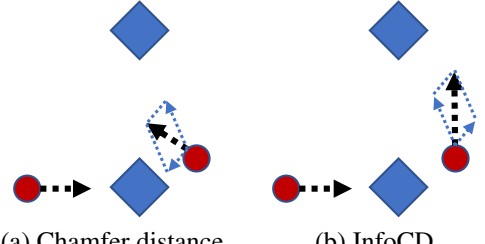

(a) Chamfer distance      (b) InfoCD

Figure 3: Illustration of the moving directions of matched points using **(a)** Chamfer distance or **(b)** InfoCD. Blue: ground truth; Red: predictions.

empirically we observe that such training stabilization can be efficiently reached using different networks on different datasets as well. For instance, as illustrated in Fig. 5(a)[3], InfoCD with L1-distance in Eq. 4 behaves similarly in terms of convergence rate to L1-CD that has convergence guarantee, but reduces the loss more significantly.

As illustrated in Fig. 5(b), ideally CD aims to reach the status where all the predicted points will be aligned perfectly with ground truth with no errors by consistently minimizing the distances. Such a requirement may be so strict that in learning the optimization may be much easier to be stuck at the suboptimal solutions, *e.g.,* forming clusters around some points. In contrast, InfoCD aims to align the point distributions with sufficiently small errors, as illustrated in Fig. 5(c) with the dotted lines indicating a Voronoi diagram for the ground-truth point cloud. This is much less strict than CD, and thus will be more likely to locate better solutions. In fact, the movements of predicted points in Fig. 3(b) provide an effective way towards learning such an alignment in Fig. 5(c) by avoiding bad suboptimal solutions.

**Choice of** $d$ **in InfoCD.** In particular, we utilize L1-distance in Eq. 4 as the choice for $d$ in InfoCD, because it is unbiased to the distance, *i.e.,* $\nabla d(x_{ij}, y_{ik}) = \nabla \|x_{ij} - y_{ik}\|$. In practice, we observe that using $d(x_{ij}, y_{ik}) = \|x_{ij} - y_{ik}\|^p, p > 0, p \neq 1$, the performance of InfoCD is very unstable for different networks

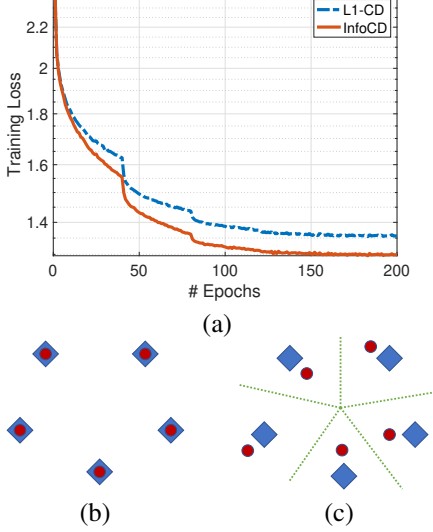

(a)

(b)          (c)

Figure 5: Illustration of **(a)** training loss using CP-Net on ShapeNet-Part, and ideal point alignment with **(b/c)** CD/InfoCD.

---

[3]The two loss curves are aligned for a better view so that the starting values are identical.

across different datasets. This is understandable: in the extreme cases where $\|x_{ij} - y_{ik}\| = 0, p > 1$ would make the predicted point unchanged, while $0 < p < 1$ would make the move of the predicted point very far away. Such updates will violate the goal of InfoCD as shown in Fig. 5(c). Other distance metrics will be investigated in our future work.

## 4 Experiments

**Datasets.** We conducted experiments for point cloud completion on the following datasets:

- *PCN [10]:* This is a subset of ShapeNet [66] with shapes from 8 categories. The incomplete point clouds are generated by back-projecting 2.5D depth images from 8 viewpoints in order to simulate real-world sensor data. For each shape, 16,384 points are uniformly sampled from the mesh surfaces as complete ground truth, and 2,048 points are sampled as partial input [10, 8].
- *Multi-view partial point cloud (MVP) [67]:* This dataset covers 16 categories with 62,400 and 41,600 pairs for training and testing, respectively. It renders the partial 3D shapes from 26 uniformly distributed camera poses for each 3D CAD model selected from ShapeNet [68], and the ground-truth point cloud is sampled via Poisson Disk Sampling (PDS).
- *ShapeNet-55/34 [16]:* ShapeNet-55 contains 55 categories in ShapeNet with 41,952 shapes for training and 10,518 shapes for testing. ShapeNet-34 uses a subset of 34 categories for training and leaves 21 unseen categories for testing where 46,765 object shapes are used for training, 3,400 for testing on seen categories and 2,305 for testing on novel (unseen) categories. In both datasets, 2,048 points are sampled as input and 8,192 points as ground truth. Following the same evaluation strategy with [16], 8 fixed viewpoints are selected and the number of points in the partial point cloud is set to 2,048, 4,096 or 6,144 (25%, 50% or 75% of a complete point cloud) which corresponds to three difficulty levels of *simple*, *moderate* and *hard* in the test stage.
- *ShapeNet-Part [65]:* This is a subset of ShapeNetCore [66] 3D meshes, containing 17,775 different 3D meshes with 16 categories. The ground-truth point clouds were created by sampling 2,048 points uniformly on each mesh. The partial point clouds were generated by randomly selecting a viewpoint as a center among multiple viewpoints and removing points within a certain radius from the complete data. The number of points we remove from each point cloud is 512.

**Implementation.** We considered three state-of-the-art networks, CP-Net [64], PointAttN [1] and Seed-Former [8], as our backbone networks for comparison and analysis. We also applied InfoCD to almost all the popular completion networks, *i.e.,* PCN [10], FoldingNet [23], TopNet [69], MSN [33], Cascaded [25], VRC [67], PMP-Net [73], PoinTr [16], SnowflakeNet [17], to verify its performance by replacing the original CD loss wherever it occurs. We performed the same replacement for all the other comparative losses in our experiments. We trained all these networks from scratch using Py-Torch, optimized by either Adam [74] or AdamW [75]. Hyperparameters such as learning rates, batch sizes and balance factors in the original losses for training baseline

Table 1: Comparison on PCN in terms of per-point L1-CD $\times 1000$.

| Methods | Plane | Cabinet | Car | Chair | Lamp | Couch | Table | Boat | Avg. |
|---|---|---|---|---|---|---|---|---|---|
| TopNet [69] | 7.61 | 13.31 | 10.90 | 13.82 | 14.44 | 14.78 | 11.22 | 11.12 | 12.15 |
| AtlasNet [70] | 6.37 | 11.94 | 10.10 | 12.06 | 12.37 | 12.99 | 10.33 | 10.61 | 10.85 |
| GRNet [71] | 6.45 | 10.37 | 9.45 | 9.41 | 7.96 | 10.51 | 8.44 | 8.04 | 8.83 |
| CRN [25] | 4.79 | 9.97 | 8.31 | 9.49 | 8.94 | 10.69 | 7.81 | 8.05 | 8.51 |
| NSFA [24] | 4.76 | 10.18 | 8.63 | 8.53 | 7.03 | 10.53 | 7.35 | 7.48 | 8.06 |
| FBNet [72] | 3.99 | 9.05 | 7.90 | 7.38 | 5.82 | 8.85 | 6.35 | 6.18 | 6.94 |
| PCN [10] | 5.50 | 22.70 | 10.63 | 8.70 | 11.00 | 11.34 | 11.68 | 8.59 | 11.27 |
| HyperCD [35] + PCN | 5.95 | **11.62** | **9.33** | 12.45 | 12.58 | 13.10 | **9.82** | 9.85 | **10.59** |
| **InfoCD + PCN** | **5.07** | 22.27 | 10.18 | **8.26** | **10.57** | **10.98** | 11.23 | **8.15** | 10.83 |
| FoldingNet [23] | 9.49 | 15.80 | 12.61 | 15.55 | 16.41 | 15.97 | 13.65 | 14.99 | 14.31 |
| HyperCD + FoldingNet | **7.89** | 12.90 | **10.67** | 14.55 | **13.87** | **14.09** | 11.86 | **10.89** | **12.09** |
| **InfoCD+FoldingNet** | 7.90 | **12.68** | 10.83 | **14.04** | 14.05 | 14.56 | **11.61** | 11.45 | 12.14 |
| PMP-Net [73] | 5.65 | 11.24 | 9.64 | 9.51 | 6.95 | 10.83 | 8.72 | 7.25 | 8.73 |
| HyperCD + PMP-Net | 5.06 | 10.67 | 9.30 | 9.11 | 6.83 | 11.01 | 8.18 | 7.03 | 8.40 |
| **InfoCD+PMP-Net** | **4.67** | **10.09** | **8.87** | **8.59** | **6.38** | **10.48** | **7.51** | **6.75** | **7.92** |
| PoinTr [16] | 4.75 | 10.47 | 8.68 | 9.39 | 7.75 | 10.93 | 7.78 | 7.29 | 8.38 |
| HyperCD + PoinTr | 4.42 | 9.77 | 8.22 | 8.22 | 6.62 | 9.62 | 6.97 | 6.67 | 7.56 |
| **InfoCD + PoinTr** | **4.06** | **9.42** | **8.11** | **7.81** | **6.21** | **9.38** | **6.57** | **6.40** | **7.24** |
| SnowflakeNet [17] | 4.29 | 9.16 | 8.08 | 7.89 | 6.07 | 9.23 | 6.55 | 6.40 | 7.21 |
| HyperCD + SnowflakeNet | **3.95** | 9.01 | 7.88 | 7.37 | **5.75** | 8.94 | **6.19** | 6.17 | 6.91 |
| **InfoCD + SnowflakeNet** | 4.01 | **8.81** | **7.62** | **5.51** | 5.80 | **8.91** | 6.21 | **5.05** | **6.86** |
| PointAttN [1] | 3.87 | 9.00 | 7.63 | 7.43 | 5.90 | 8.68 | 6.32 | 6.09 | 6.86 |
| HyperCD + PointAttN | 3.76 | 8.93 | 7.49 | 7.06 | 5.61 | 8.48 | 6.25 | **5.92** | 6.68 |
| **InfoCD + PointAttN** | **3.72** | **8.87** | **7.46** | **7.02** | **5.60** | **8.45** | **6.23** | **5.92** | **6.65** |
| SeedFormer [8] | 3.85 | 9.05 | 8.06 | 7.06 | 5.21 | 8.85 | 6.05 | 5.85 | 6.74 |
| HyperCD + SeedFormer | 3.72 | **8.71** | 7.79 | **6.83** | 5.11 | **8.61** | **5.82** | 5.76 | 6.54 |
| **InfoCD + SeedFormer** | **3.69** | 8.72 | **7.68** | 6.84 | **5.08** | **8.61** | 5.83 | **5.75** | **6.52** |

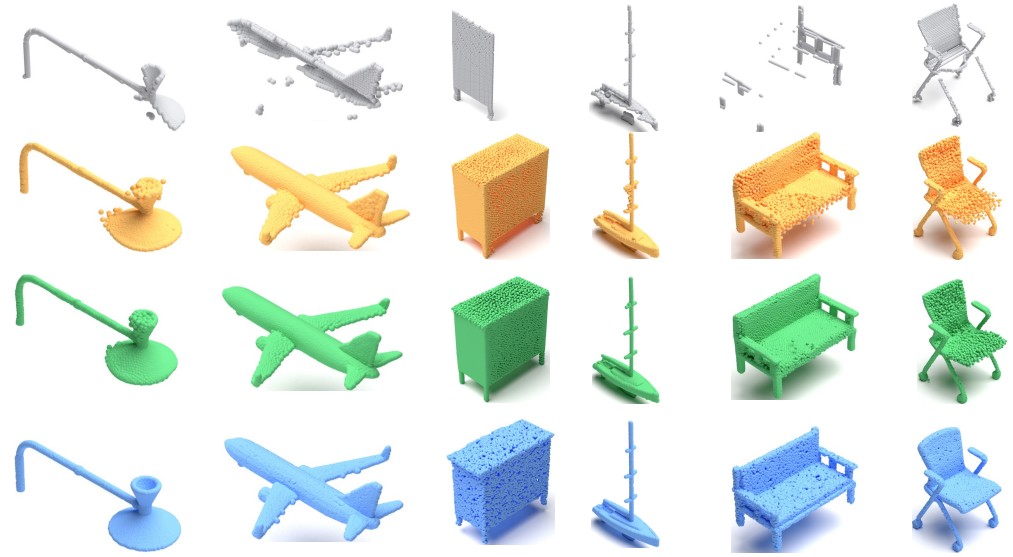

Figure 6: Visual result comparison on PCN. **Row-1:** Inputs of incomplete point clouds. **Row-2:** Outputs of Seedformer with CD. **Row-3:** Outputs of Seedformer with InfoCD. **Row-4:** Ground truth.

Table 2: Completion results on MVP in terms of L2-CD $\times 10^4$ and EMD $\times 10^2$.

| | Methods | airplane | cabinet | car | chair | lamp | sofa | tabel | watercraft | bed | bench | bookshelf | bus | guitar | motorbike | pistol | skateboard | Avg. |
|---|---|---|---|---|---|---|---|---|---|---|---|---|---|---|---|---|---|---|
| CD | PCN [10] | 4.50 | 8.83 | 6.41 | 13.01 | 21.33 | 9.90 | 12.86 | 9.46 | 20.00 | 10.26 | 14.63 | 4.94 | 1.73 | 6.17 | 5.84 | 5.76 | 9.78 |
| | **InfoCD+PCN** | **3.95** | **8.82** | **6.38** | **12.03** | **17.43** | **9.63** | **12.41** | **8.69** | **18.92** | **8.75** | **13.40** | **5.02** | **1.84** | **6.06** | **5.81** | **4.37** | **9.41** |
| | TopNet [69] | 4.12 | 9.84 | 7.44 | 13.26 | 18.64 | 10.77 | 12.95 | 8.98 | 19.99 | 9.21 | 16.06 | 5.47 | 2.36 | 7.06 | 7.04 | 4.68 | 10.30 |
| | **InfoCD+TopNet** | **3.98** | **9.81** | **7.42** | **13.24** | **17.87** | **10.52** | **12.45** | **8.93** | **19.69** | **8.52** | **14.62** | **5.42** | **2.35** | **7.05** | **6.52** | **4.21** | **10.01** |
| | MSN [33] | 2.73 | 8.92 | 6.50 | 10.75 | 13.37 | 9.26 | 10.17 | 7.70 | 17.27 | 6.64 | 12.10 | 5.21 | 1.37 | 4.59 | 4.62 | 3.38 | 7.99 |
| | **InfoCD+MSN** | **7.28** | **8.51** | **6.03** | **10.18** | **12.91** | **8.87** | **9.72** | **7.24** | **16.82** | **6.21** | **11.67** | **4.79** | **0.91** | **4.15** | **4.17** | **2.97** | **7.56** |
| | Cascaded [25] | 2.54 | 8.62 | 5.93 | 8.76 | 11.22 | 8.46 | 9.20 | 6.61 | 14.63 | 6.09 | 10.17 | 4.95 | 1.55 | 4.34 | 4.23 | 3.19 | 7.25 |
| | **InfoCD+Cascaded** | **2.43** | **8.05** | **5.73** | **8.77** | **10.47** | **8.24** | **9.18** | **6.41** | **14.37** | **6.02** | **10.45** | **4.70** | **1.45** | **4.23** | **4.16** | **2.99** | **7.12** |
| | VRC [67] | 2.20 | 7.92 | 5.60 | 7.49 | 8.15 | 7.45 | 7.52 | 5.20 | 11.90 | 4.88 | 7.39 | 4.53 | 1.15 | 3.90 | 3.44 | 3.22 | 6.09 |
| | **InfoCD+VRC** | **2.03** | **7.88** | **5.41** | **7.31** | **7.92** | **7.22** | **7.30** | **5.01** | **11.67** | **4.65** | **7.14** | **4.30** | **0.97** | **4.68** | **3.19** | **3.04** | **5.87** |
| EMD | PCN | 4.70 | 7.99 | 5.75 | 6.90 | 11.99 | 5.32 | 6.60 | 5.40 | 9.84 | 4.85 | 7.87 | 5.24 | 10.56 | 4.93 | 4.86 | 5.59 | 6.80 |
| | **InfoCD+PCN** | **3.75** | **5.59** | **3.97** | **5.23** | **10.11** | **4.42** | **5.45** | **4.67** | **7.29** | **4.21** | **5.55** | **3.53** | **6.12** | **4.02** | **4.70** | **3.84** | **5.17** |
| | TopNet | 4.89 | 6.30 | 4.07 | 7.01 | 10.75 | 6.47 | 7.50 | 4.68 | 8.09 | 6.27 | 6.80 | 3.50 | 4.21 | 4.26 | 6.02 | 3.49 | 6.18 |
| | **InfoCD+TopNet** | **4.47** | **6.02** | **3.81** | **6.82** | **10.21** | **6.05** | **7.12** | **4.37** | **7.87** | **5.87** | **6.02** | **3.31** | **4.06** | **4.11** | **5.82** | **3.15** | **5.72** |
| | MSN | 2.75 | 4.02 | 3.47 | 4.44 | 6.28 | 3.74 | 4.46 | 3.82 | 5.27 | 3.34 | 4.28 | 2.92 | 2.07 | 3.30 | 3.62 | 2.21 | 3.94 |
| | **InfoCD+MSN** | **2.18** | **3.51** | **2.97** | **3.96** | **5.77** | **3.21** | **3.92** | **3.24** | **4.75** | **2.86** | **3.79** | **2.41** | **1.50** | **2.81** | **3.09** | **2.64** | **3.38** |
| | Cascaded | 3.03 | 6.82 | 5.44 | 5.16 | 7.55 | 5.57 | 4.73 | 4.88 | 6.85 | 3.51 | 5.71 | 5.81 | 5.30 | 4.30 | 4.42 | 3.44 | 5.18 |
| | **InfoCD+Cascaded** | **2.87** | **6.23** | **5.39** | **5.06** | **7.10** | **5.45** | **4.57** | **4.79** | **6.42** | **3.49** | **5.15** | **5.72** | **3.58** | **4.19** | **4.27** | **2.91** | **5.01** |
| | VRC | 3.03 | 7.57 | 6.14 | 5.49 | 6.15 | 5.80 | 4.65 | 4.97 | 6.58 | 3.45 | 5.28 | 6.59 | 3.08 | 4.45 | 4.56 | 3.20 | 5.27 |
| | **InfoCD+VRC** | **2.68** | **7.26** | **5.83** | **5.15** | **5.82** | **5.49** | **4.36** | **4.68** | **6.22** | **3.13** | **4.97** | **6.26** | **2.77** | **4.13** | **4.15** | **2.89** | **4.97** |

networks were kept consistent with the baseline settings for fair comparisons. Hyperparameter $\tau$ in InfoCD was tuned based on grid search, while $\lambda$ was set to $10^{-7}$ for all the experiments. We conducted our experiments on a server with 4 NVIDIA A100 80G GPUs and one with 10 NVIDIA Quadro RTX 6000 24G GPUs due to the large model sizes of some baseline networks.

**Evaluation.** We evaluated the best performance of all the methods using CD (lower is better). We also used F1-Score@1% [76] (higher is better) to evaluate the performance. Note that for the PCN method, we report the results based on the PyTorch implementation from PMP-Net. When using the original TensorFlow implementation, it achieves an average result of 10.28 on PCN with the default hyperparameters. However, when trained with our infoCD, it can achieve a lower value of 9.86.

## 4.1 State-of-the-art Comparison

**PCN.** Following the literature, we report CD with L1-distance in Table 1. As we can see, InfoCD is able to improve the performance of all the baselines consistently and significantly, achieving new

state-of-the-art results. As discussed earlier, numerical metrics such as CD may not faithfully reflect the visual quality, we also provide qualitative evaluation results in Fig. 6, compared with results generated from Seedformer trained with the CD loss. As we can see, both models can reconstruct point clouds in general outlines to some extent, but the completion results with CD are more likely to suffer from distortion in several areas with high noise levels on the surface. In contrast, InfoCD can help the baseline network demonstrably better reconstruct point clouds in general outlines while maintaining the realistic details of the original ground truth with significant noise reduction.

**MVP.** We evaluate the universality of InfoCD on this dataset with several popular networks for tasks with higher diversities and different designs of architectures. Following the literature, we report CD with L2-distance and EMD in Table 2. Similar to Table 1, the introduction of InfoCD can consistently and significantly improve the performance of all different baselines in both two metrics.

**ShapeNet-55/34.** We evaluate the adaptability of InfoCD on both datasets for the tasks with higher diversities. Table 3 lists the L2-CD on three difficulty levels as well as the average. Following the literature, we show the results in 5 categories (Table, Chair, Plane, Car, and Sofa) whose numbers of training samples are more than 2,500. We also provide the results using the

Table 3: Results on ShapeNet-55 using L2-CD$\times$1000 and F1 score.

| Methods | Table | Chair | Plane | Car | Sofa | CD-S | CD-M | CD-H | Avg. | F1 |
|---|---|---|---|---|---|---|---|---|---|---|
| PFNet [77] | 3.95 | 4.24 | 1.81 | 2.53 | 3.34 | 3.83 | 3.87 | 7.97 | 5.22 | 0.339 |
| TopNet [69] | 2.21 | 2.53 | 1.14 | 2.18 | 2.36 | 2.26 | 2.16 | 4.3 | 2.91 | 0.126 |
| PCN [10] | 2.13 | 2.29 | 1.02 | 1.85 | 2.06 | 1.94 | 1.96 | 4.08 | 2.66 | 0.133 |
| GRNet [71] | 1.63 | 1.88 | 1.02 | 1.64 | 1.72 | 1.35 | 1.71 | 2.85 | 1.97 | 0.238 |
| FoldingNet [23] | 2.53 | 2.81 | 1.43 | 1.98 | 2.48 | 2.67 | 2.66 | 4.05 | 3.12 | 0.082 |
| **InfoCD + FoldingNet** | **2.14** | **2.37** | **1.03** | **1.55** | **2.04** | **2.17** | **2.50** | **3.46** | **2.71** | **0.137** |
| PoinTr [16] | 0.81 | 0.95 | 0.44 | 0.91 | 0.79 | 0.58 | 0.88 | 1.79 | 1.09 | 0.464 |
| **InfoCD + PoinTr** | **0.69** | **0.83** | **0.33** | **0.80** | **0.67** | **0.47** | **0.73** | **1.50** | **0.90** | **0.524** |
| SeedFormer [8] | 0.72 | 0.81 | 0.40 | 0.89 | 0.71 | 0.50 | 0.77 | 1.49 | 0.92 | 0.472 |
| HyperCD + SeedFormer | 0.66 | 0.74 | 0.35 | 0.83 | 0.64 | 0.47 | 0.72 | 1.40 | 0.86 | 0.482 |
| **InfoCD + SeedFormer** | **0.65** | **0.72** | **0.31** | **0.81** | **0.62** | **0.43** | **0.71** | **1.38** | **0.84** | **0.490** |

F1 metric. Again InfoCD has significantly improved the baseline models, especially when networks are simpler such as FoldingNet. Please refer to our supplementary for more qualitative evaluations.

On ShapeNet-34, we evaluate performances within 34 seen categories (same as training) as well as 21 unseen categories (not used in training) and list our results in Table 4. We can observe that, again, InfoCD can improve the performance of baseline models, indicating that InfoCD is highly generalizable for point cloud completion.

Table 4: Results on ShapeNet-34 using L2-CD$\times$1000 and F1 score.

| Methods | 34 seen categories | | | | | 21 unseen categories | | | | |
|---|---|---|---|---|---|---|---|---|---|---|
| | CD-S | CD-M | CD-H | Avg. | F1 | CD-S | CD-M | CD-H | Avg. | F1 |
| PFNet [77] | 3.16 | 3.19 | 7.71 | 4.68 | 0.347 | 5.29 | 5.87 | 13.33 | 8.16 | 0.322 |
| TopNet [69] | 1.77 | 1.61 | 3.54 | 2.31 | 0.171 | 2.62 | 2.43 | 5.44 | 3.50 | 0.121 |
| PCN [10] | 1.87 | 1.81 | 2.97 | 2.22 | 0.154 | 3.17 | 3.08 | 5.29 | 3.85 | 0.101 |
| GRNet [71] | 1.26 | 1.39 | 2.57 | 1.74 | 0.251 | 1.85 | 2.25 | 4.87 | 2.99 | 0.216 |
| FoldingNet [23] | 1.86 | 1.81 | 3.38 | 2.35 | 0.139 | 2.76 | 2.74 | 5.36 | 3.62 | 0.095 |
| **InfoCD + FoldingNet** | **1.54** | **1.60** | **3.10** | **2.08** | **0.177** | **2.42** | **2.49** | **5.01** | **3.31** | **0.157** |
| PoinTr [16] | 0.76 | 1.05 | 1.88 | 1.23 | 0.421 | 1.04 | 1.67 | 3.44 | 2.05 | 0.384 |
| **InfoCD + PoinTr** | **0.47** | **0.69** | **1.35** | **0.84** | **0.529** | **0.61** | **1.06** | **2.55** | **1.41** | **0.493** |
| SeedFormer [8] | 0.48 | 0.70 | 1.30 | 0.83 | 0.452 | 0.61 | 1.08 | 2.37 | 1.35 | 0.402 |
| HyperCD + SeedFormer | 0.46 | 0.67 | 1.24 | 0.79 | 0.459 | 0.58 | 1.03 | 2.24 | 1.31 | 0.428 |
| **InfoCD + SeedFormer** | **0.43** | **0.63** | **1.21** | **0.75** | **0.581** | **0.54** | **1.01** | **2.18** | **1.24** | **0.449** |

**ShapeNet-Part.** Previous results in Figs. 4 and 5(a) show the performance of CP-Net on this dataset. Below we summarize more results: Table 5 lists our comparison results of CP-Net trained with some popular losses in the forms of mean and standard deviation (std). From the perspective of the mean, it is clear that InfoCD outperforms the others. Moreover, from the perspective of the std, training with InfoCD is much more stable than the others. Fig. 7 shows the ablation study for learning rate ($lr$) and $\tau$ in Eq. 5 in terms of L2-CD. Overall, training with InfoCD is not sensitive to such hyperparameters.

Table 5: CP-Net Avg. results on ShapeNet-Part.

| Loss | L2-CD$*10^3$ |
|---|---|
| L1-CD | 4.16$\pm$0.028 |
| L2-CD | 4.82$\pm$0.117 |
| DCD [20] | 5.74$\pm$0.049 |
| HyperCD | 4.03 $\pm$0.007 |
| **InfoCD** | **4.01$\pm$0.004** |

## 5 Conclusion

In this paper, we propose a new loss for point cloud completion, namely InfoCD, a contrastive Chamfer distance loss. We show that by regularizing the CD loss with contrastive learning, InfoCD can better align the point distributions between prediction and ground truth, achieving the goal of

measuring the underlying geometric surfaces of the point clouds using mutual information (MI) estimation. In particular, we discuss and analyze the relations between InfoCD and MI, the moves of predicted points in learning, training convergence, loss landscapes, and the choice of distance metric in InfoCD. Comprehensive experiments have been conducted to demonstrate its effectiveness and efficiency using 7 networks on 5 datasets, leading to new state-of-the-art results.

**Limitations.** Due to the introduction of a new hyper-parameter $\tau$ in InfoCD, tuning hyperparameter based on grid search may need more effort, as illustrated in Fig. 7. Also due to the higher nonconvexity of In-foCD than CD, it may take more time (or epochs) to train complicated networks (*e.g.,* with large numbers of parameters, or architectures) with InfoCD.

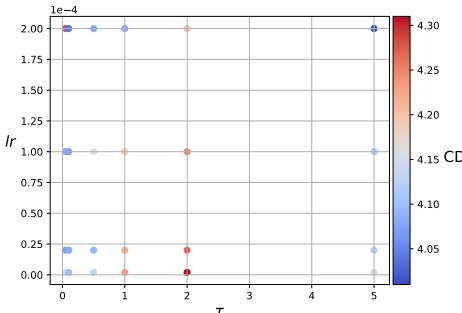

Figure 7: Ablation study on $lr$ *vs.* $\tau$.

## Acknowledgement

Fangzhou Lin was supported in part by the Support for Pioneering Research Initiated by the Next Generation (SPRING) from the Japan Science and Technology Agency. Yun Yue and Ziming Zhang were supported partially by NSF CCF-2006738. Dr. Kazunori D Yamada was supported in part by the Top Global University Project from the Ministry of Education, Culture, Sports, Science, and Technology of Japan (MEXT). Vijaya B Kolachalama was supported by the National Institutes of Health (R01-HL159620, R21-CA253498, R43-DK134273, RF1-AG062109, and P30-AG073104), the American Heart Association (20SFRN35460031), and the Karen Toffler Charitable Trust. Venkatesh Saligrama was supported by the Army Research Office Grant W911NF2110246, AFRLGrant FA8650-22-C1039, the National Science Foundation grants CCF2007350 and CCF-1955981. Computations were partially performed on the NIG supercomputer at ROIS National Institute of Genetics.

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
