# OpenReview forum: "InfoCD: A Contrastive Chamfer Distance Loss for Point Cloud Completion"
_NeurIPS.cc/2023/Conference — NeurIPS 2023 poster_

### Official Review · Reviewer_hcgP · 2023-07-02

**Soundness:** 3 good
**Presentation:** 3 good
**Contribution:** 3 good
**Rating:** 6
**Confidence:** 4

**Summary:**

The paper proposes a contrastive chamfer distance (InfoCD) for point cloud completion. More specifically, the paper shows that minimizing InfoCD is equivalent to maximizing a lower bound of the mutual information between the underlying geometric surfaces, which plays a crucial role in generating and reconstructing detailed object shapes . To verify the effectiveness of the method, extensive experiments are conducted and promising results are obtained.

**Strengths:**

1. The idea is interesting and the paper is well-organized.
2. Extensive experiments are conducted and performances are promising.

**Weaknesses:**

1. Since the real-world scenario is a critical application in point cloud completion, how about the results on the real-world dataset such as KITTI shown in existing works?
2. The evaluation metric in Tab.1 should be L2-CD as well according to the PCN paper, since a square root is calculated in the evaluation code.
3. It is worth comparing the time and memory efficiency among different methods, especially when they are applied to real-world applications or mobile devices.
4. Inconsistency citation names for [25].

**Questions:**

None.

**Limitations:**

Yes, the authors adequately addressed the limitations.

---

> ### Author Rebuttal · Authors · 2023-08-08
>
> We sincerely thank the reviewer for the valuable comments. Below are our responses to the questions arising in the review:
>
> **1. Results on KITTI:** Following GRNet (*Xie et. al. "Grnet: Gridding residual network for dense point cloud completion". In ECCV, 2020.*), we take a sequence of Lidar scans from KITTI. We then (1) extract points per frame within the object bounding boxes labeled as cars, (2) transform these incomplete point clouds to the box's coordinates, (3) complete them with a model pre-trained on cars from ShapeNet, and (4) finally transform the outputs back to the world coordinates. The table below lists our results where, again, InfoCD can improve the baselines consistently ("Fidelity" and "MMD" are two distance metrics, the smaller, the better. Please refer to GRNet for more details), demonstrating that InfoCD can generalize well for different datasets (together with the other 4 datasets in the paper). Note that here we still use the default hyperparameters in the original code for fair comparisons.
>
> --------------------
>
> Method | Fidelity$\downarrow$ | MMD$\downarrow$
>
> --------------------
>
> FoldingNet | 7.467 | 0.537
>
> **InfoCD+FoldingNet** | **1.944** | **0.333**
>
> ---------------------
>
> PoinTr | 0.000 | 0.526
>
> **InfoCD+PoinTr** | 0.000 | **0.502**
>
> --------------------
>
> **2. Evaluation metric in Table 1:** We follow the paper *Yu et. al. "PoinTr: Diverse Point Cloud Completion with Geometry-Aware Transformers". In ICCV, 2021.*, and use their evaluation code where L1-CD is used. This is different from the PCN paper.
>
> **3. Training time and GPU memory footprint for computational efficiency:** InfoCD has only a few more operations than CD and thus in theory both computational efficiency should be similar. Numerically, for training CP-Net with CD and InfoCD per iteration it takes 0.4239$\pm$0.0019 and **0.4498$\pm$0.0030** second with 1052.627$\pm$0.0374 and **1053.692$\pm$0.0425** MB in GPU memory, respectively.
>
> **4. We will check the references to make everything consistent.**

---

> > ### Comment · Reviewer_hcgP · 2023-08-14
> >
> > Thanks for the authors' rebuttal.
> >
> > If the evaluation code is different from the original PCN paper, why are the avg. metric of PCN shown in Tab.1 is the same as the original PCN paper?
> >
> > Moreover, after double-checking the avg. metric, it seems that the avg. value 9.64 is not equal to the average value of all the eight classes (while it is not for PCN paper), which actually should not be the case since PCN dataset has the equal number of testing data per category.
> >
> > Is there any reason for this, and could the authors double-check the evaluation?

---

> > > ### Author Response · Authors · 2023-08-14
> > > **Thanks for your questions!**
> > >
> > > We use the public code with the paper "PoinTr: Diverse Point Cloud Completion with Geometry-Aware Transformers" in ICCV 2021. Please refer to https://github.com/yuxumin/PoinTr/tree/master where the evaluation code is located at https://github.com/yuxumin/PoinTr/blob/master/tools/runner.py
> > >
> > > In fact, there are some other papers citing **exactly the same numbers** on PCN as PoinTr and ours. For instance,
> > >
> > > 1. Wen et. al. "Pmp-net: Point cloud completion by learning multi-step point moving paths." In CVPR 2021. (See Table 2)
> > >
> > > 2. Xiang et. al. "SnowflakeNet: Point cloud completion by snowflake point deconvolution with skip-transformer." In ICCV 2021. (See Table 1)
> > >
> > > 3. Zhou et. al. "SeedFormer: Patch Seeds based Point Cloud Completion with Upsample Transformer." In ECCV 2022. (See Table 1)
> > >
> > > We have tested the code, with **no change** on the evaluation, and got similar numbers. For consistency with other papers, we decided to copy these numbers into our table. We encourage the reviewer to test the code above for verification.
> > >
> > > --------------------------
> > >
> > > **WE NOW FOUND A BUG IN THE EVALUATION CODE THAT WAS RELEASED ONE AND HALF YEARS AGO!!! THANKS A LOT FOR THE REVIEWER'S COMMENTS!!!**
> > >
> > > The code we are using was released one and half years ago, where we found that the category "Cabinet" was not added into the calculation for average. This is due to a bug in the test json file where "Cabinet" should use "c" instead of "C". After correcting the bug, the average numbers for PCN and ours are 11.27 and 10.84, respectively. Ours is still better. We will correct the numbers in the table as well.

---

> > > > ### Comment · Reviewer_hcgP · 2023-08-15
> > > >
> > > > It was great that the authors found the reason for this. Besides re-evaluating PCN, it is highly encouraged to re-evaluate other methods in Tab.1 for a fair comparison.

---

> > > > > ### Author Response · Authors · 2023-08-15
> > > > >
> > > > > Thank you so much for the suggestions. We have checked the rest results in Table 1 and they are correct. The bug is only applied to PCN evaluation.
> > > > >
> > > > > More specifically, for each method, we tried to run experiments with their own authors’ public code for fair comparison. Only PCN and foldingnet are using PoinTr’ code (for foldingnet, the original paper didn’t provide point cloud completion task). And in order to avoid misuse of different methods, we run each method in different projects. That’s why this bug only occurs in PCN evaluation (which we are using a relatively old version of their code).

---

> > > > > > ### Comment · Reviewer_hcgP · 2023-08-17
> > > > > >
> > > > > > It is obvious that PCN is much earlier than PointTr, thus, why the authors using PointTr's code to run PCN work instead of directly using original PCN's public code?
> > > > > >
> > > > > > Moreover, after checking [seedformer's code](https://github.com/hrzhou2/seedformer/blob/master/codes/manager.py#L306), we can see that actually they are also using sqrt distance instead of L1 even though they write L1 in their paper, that's why they get consistent metric with the original PCN paper. However, the authors got different values compared to PCN (11.27 vs 9.64) because of L1 distance while still comparing to other works with sqrt metric, this hurts the validity of the work and makes the results less convincing.

---

> > > > > > > ### Author Response · Authors · 2023-08-17
> > > > > > > **Clarification on Reviewer Concerns**
> > > > > > >
> > > > > > > Our previous response appears to have been misunderstood. Let us clarify through a longer response. \
> > > > > > > **TLDR:** We stand by our comparisons and insist that they are fair, and our results are valid and convincing.
> > > > > > >
> > > > > > > 1. *Reviewer: Checking seedformer's code, we can see that actually they are also using sqrt distance instead of L1 even though they write L1 in their paper, that's why they get consistent metric with the original PCN paper.*
> > > > > > >
> > > > > > > We believe that the reviewer has **misunderstood** our notation. We are using the same distance metric as seedformer. Please refer to **our definition of L1 CD in Eq. 4**. What we call L1 distance is the **Euclidean** distance, which is the **square root** distance that reviewer mentions. The Euclidean distance is the square root of the sum of squares of differences between the components of $x_{ij}$ and $y_{ik}$. This can also be inferred from the fact that what we call L2 distance is the Euclidean distance squared.
> > > > > > >
> > > > > > > 2. *Reviewer: the authors got different values compared to PCN (11.27 vs 9.64) because of L1 distance while still comparing to other works with sqrt metric, this hurts the validity of the work and makes the results less convincing.*
> > > > > > >
> > > > > > > This is not **true**. Fundamentally, the original PoinTr's code for PCN calculation has a **bug.** Specifically, PoinTr perhaps by oversight overlooks the contribution from Cabinet class in its PCN evaluation. Indeed upon excluding Cabinet class, which contributes a value of 22.70, PCN results in the average: **9.64**=(5.50+10.63+8.7+11+11.34+11.68+8.59)/7. Including Cabinet class, PCN achieves the average **11.27**=(5.50+22.70+10.63+8.7+11+11.34+11.68+8.59)/8. **This same mistake/oversight appears in later works that report PCN**.\
> > > > > > > That said our InfoCD approach yields **9.20** = (5.07+10.18+8.26+10.57+10.98+11.23+8.15)/7 if Cabinet is **excluded**.
> > > > > > > InfoCD results in **10.83**= (5.07+22.27+10.18+8.26+10.57+10.98+11.23+8.15)/8 if Cabinet class is **included**.
> > > > > > >
> > > > > > > 3.  *Reviewer: Why not using the PCN’s code directly?*
> > > > > > >
> > > > > > > First, all the recent works (e.g., Pmp-net, SnowflakeNet, SeedFormer) report the same numbers as produced by the PoinTr’s code. Second, PCN’s code was written using TensorFlow, but all other comparison methods were written using Pytorch, and our code is using Pytorch as well.  For fair comparison, we use the PoinTr’s code.

---

> > > > > > > > ### Comment · Reviewer_mFMW · 2023-08-18
> > > > > > > > **overlooks the contribution from Cabinet**
> > > > > > > >
> > > > > > > > Indeed, the value 9.64 originates from PMP-Net, and it became publicly available on arXiv in December 2020. Both PoinTr and SnowflakeNet are concurrent work that reference the aforementioned value from PMP-Net.

---

> > > > > > > > > ### Author Response · Authors · 2023-08-18
> > > > > > > > >
> > > > > > > > > We thank the reviewer for pointing this out.

---

> > > > > > > > > > ### Comment · Reviewer_hcgP · 2023-08-19
> > > > > > > > > >
> > > > > > > > > > I am appreciating the effectiveness of the work on the other datasets, but it is still unreasonable that subsequent works get different evaluation values for PCN if they are using the same testing data and evaluation method (both are sqrt version) with PCN. Moreover, Tensorflow should not be the issue of not using PCN's original code and at least very close values should be obtained regardless of deep learning tools.
> > > > > > > > > >
> > > > > > > > > > Hence, I would like to slightly lower my rating but still be positive on this work.

---

> > > > > > > > > > > ### Author Response · Authors · 2023-08-19
> > > > > > > > > > > **Lower rating but positive**
> > > > > > > > > > >
> > > > > > > > > > > While we are happy with reviewer maintaining a positive rating, it appears that we did not get through.
> > > > > > > > > > >
> > > > > > > > > > > **Once again the reason PMP-NET, PoinTr. SnowflakeNet, SeedFormer have 9.64 is because they are all consistently incorrect in reporting PCN. All of these papers overlooked contribution from Cabinet class.**
> > > > > > > > > > >
> > > > > > > > > > > Finally, the point is less about Tensorflow vs. Pytorch in PoinTr's code, but more about the fact that all the other subsequent papers report the same numbers with Pytorch code. **So the only way to fairly compare with such recent works is to use the same Pytorch code for training and evaluation.**

---

> > > > > > > > > > > > ### Author Response · Authors · 2023-08-20
> > > > > > > > > > > >
> > > > > > > > > > > > Dear Reviewer hcgP,
> > > > > > > > > > > >
> > > > > > > > > > > > We have identified one potential reason on why the original PCN’s performance is better than the concurrent works. The number of input points per point cloud for the original PCN is **3000**, while for others the number is 2048. Please refer to parser.add_argument('--num_input_points', type=int, default=3000) in L155 from the file https://github.com/wentaoyuan/pcn/blob/master/train.py
> > > > > > > > > > > >
> > > > > > > > > > > > This number is crucial for the completion performance. Now we are re-running experiments using PyTorch code to see if we can achieve similar numbers with 3000 input points. We will update the performance here if time allows.

---

> > > > > > > > > > > > > ### Comment · Reviewer_hcgP · 2023-08-21
> > > > > > > > > > > > >
> > > > > > > > > > > > > I believe it is not because of the number of input points for PCN, and 2048 points provides the same results to the original PCN paper. 3000 is just an example number when running the code.
> > > > > > > > > > > > >
> > > > > > > > > > > > > Again, either Tensorflow or PyTorch should not be the reason for different results, and there are also online PyTorch PCN models which can provide similar results with original Tensorflow's, and at least it is easy to implement a PyTorch one as well.
> > > > > > > > > > > > >
> > > > > > > > > > > > > Hence, what the authors do seem unfair to PCN.

---

### Official Review · Reviewer_SCvJ · 2023-07-04

**Soundness:** 2 fair
**Presentation:** 2 fair
**Contribution:** 2 fair
**Rating:** 5
**Confidence:** 4

**Summary:**

This paper proposes a contrastive Chamfer distance loss, which introduces contrastive learning into the CD loss. Experiments are conducted on PCN, MVP, ShapeNet-55/34 and ShapeNet-Part datasets, and state-of-the-art results are achieved on these datasets.

**Strengths:**

1. The idea seems reasonable and the overall performance is good.
2. The paper is overall well written and easy to follow.


**Weaknesses:**

1. Since the proposed loss is a supervised learning loss, I am a bit worried about the generalization ability of the proposed method in different datasets.

**Questions:**

When you compare with the original methods, have you retrained these learning-based models on the same datasets?

**Limitations:**

Yes

---

> ### Author Rebuttal · Authors · 2023-08-07
>
> We sincerely thank the reviewer for the valuable comments. Below are our responses to the questions arising in the review:
>
> **Generalization ability on different datasets:** To answer this question within limited time, we test our method on **KITTI**, a real-world dataset. Following GRNet (*Xie et. al. "Grnet: Gridding residual network for dense point cloud completion". In ECCV, 2020.*), we take a sequence of Lidar scans from KITTI. We then (1) extract points per frame within the object bounding boxes labeled as cars, (2) transform these incomplete point clouds to the box's coordinates, (3) complete them with a model pre-trained on cars from ShapeNet, and (4) finally transform the outputs back to the world coordinates. The table below lists our results where, again, InfoCD can improve the baselines consistently ("Fidelity" and "MMD" are two distance metrics, the smaller, the better. Please refer to GRNet for more details), demonstrating that InfoCD can generalize well for different datasets (together with the other 4 datasets in the paper). Note that here we still use the default hyperparameters in the original code for fair comparisons.
>
> --------------------
>
> Method | Fidelity$\downarrow$ | MMD$\downarrow$
>
> --------------------
>
> FoldingNet | 7.467 | 0.537
>
> **InfoCD+FoldingNet** | **1.944** | **0.333**
>
> ---------------------
>
> PoinTr | 0.000 | 0.526
>
> **InfoCD+PoinTr** | 0.000 | **0.502**
>
> --------------------

---

> > ### Author Response · Authors · 2023-08-18
> >
> > Dear Reviewer SCvJ,
> >
> > Thanks for your valuable comments. We hope that our replies have well addressed your concerns about our submission. Please do let us know if you have more questions, and we will try to answer your questions asap. Thanks

---

> > ### Comment · Reviewer_SCvJ · 2023-08-19
> >
> > Thanks for the authors' rebuttal.
> > The author didn't answer my question about whether to retrain the models of the original methods.

---

> > > ### Author Response · Authors · 2023-08-19
> > >
> > > Thanks for your concern! **Yes**, we have retrained the models of the original methods to guarantee the numbers can be reproducible.

---

### Official Review · Reviewer_eAft · 2023-07-06

**Soundness:** 4 excellent
**Presentation:** 4 excellent
**Contribution:** 4 excellent
**Rating:** 9
**Confidence:** 5

**Summary:**

This paper proposed a novel metric to measure the similarity between two point sets, which is based on the basic formula of InfoNCE loss and the Chamfer distances. The key idea is to implicitly estimate the MI between the two point sets, and the way to achieve such target is to treat the distance of between points as a measurement of positive and negative samples.

**Strengths:**

1. The reviewer is highly in favor of this paper, as this draft addresses a very fundamental problem in the deep learning of point cloud data, which is the similarity measurement between two point sets.
2. The geometric based CD/EMD metric have been used for years in point cloud completion/reconstruction area, and the proposed InfoCD loss takes one step further to incorporate the idea of mutual information. The formulation of InfoCD, which is the combination of InfoNCE and Chamfer distance, technically makes sense and is very easy to follow.
3. The experiments are great, which covers most of the recent work and almost all popular benchmark in point cloud completion. The improvement achieved by InfoCD is non-trivial, and the generalization ability and the performance gain is also impressive.
4. Potential performance gain on a lot of related tasks such as 2D-3D reconstruction, unsupervised learning, shape generation can benefit from this work. The potential application of InfoCD may not be limited in point cloud completion task.


**Weaknesses:**

1. The convergence analysis is relatively weak, as only the experimental proves is provided instead of a more mathematical proof. This does not trouble the reviewer a lot, because the experimental result compared with CD loss looks very good and convincing.
2. It is a little bit pity that only the point cloud completion task is discussed. Maybe one or two applications on other tasks could provide more evidence on the generalization ability and the effectiveness of the InfoCD loss. For example, 2D-3D reconstruction.

In all, the reviewer does not see much weakness in this draft. It is a high-quality paper in terms of the point cloud completion research.


**Questions:**

Please see weakness.

**Limitations:**

The author has fully addressed the limitations in the draft.

---

> ### Author Rebuttal · Authors · 2023-08-07
>
> We sincerely thank the reviewer for the valuable comments. Below are our responses to the questions arising in the review:
>
> **1. Convergence:** We thank the reviewer for understanding. We will try to develop a convergence theory in our future work.
>
> **2. Generalization ability on new tasks:** Given the limited time, we add a new task of Single-View Reconstruction (SVR) that aims to reconstruct a point cloud from an image of the underlying object. Following 3DAttriFlow (*Wen et. al. "3D shape reconstruction from 2D images with disentangled attribute flow". In CVPR, 2022.*) and SnowflakeNet (*Xiang et. al. "Snowflake Point Deconvolution for Point Cloud Completion and Generation with Skip-Transformer". TPAMI, 2022.*), we sample 30k points from the watertight mesh in ShapeNet as the ground truth, and output 2048 points for evaluation based on per-point L1-CD$\times10^2$. We replace CD in SnowflakeNet with InfoCD for training, and list average comparison results below, demonstrating that InfoCD can generalize well for different tasks.
>
> -----------------------------
> Method                | Ave.
>
> -----------------------------
>
> 3DAttriFlow           | 3.02
>
> SnowflakeNet          | 2.86
>
> **InfoCD + SnowflakeNet | 2.73**
>
> -----------------------------

---

> > ### Comment · Reviewer_eAft · 2023-08-17
> >
> > Good work. I have no further question and would like to remain my ratings.

---

> > > ### Author Response · Authors · 2023-08-17
> > >
> > > Thanks a lot for your support!

---

### Official Review · Reviewer_mFMW · 2023-07-06

**Soundness:** 3 good
**Presentation:** 3 good
**Contribution:** 3 good
**Rating:** 5
**Confidence:** 5

**Summary:**

The paper introduces a novel loss function called InfoCD for point cloud completion tasks. InfoCD maximizes a lower bound of the mutual information, aiming to improve the quality of the completed point clouds. The experimental results presented in the paper demonstrate promising outcomes, indicating the effectiveness of the proposed approach.

**Strengths:**

- Exploring the improved CD loss as a research direction for point cloud reconstruction shows promise and holds significant potential.
- The paper is well-written and effectively communicates its ideas, making it easy to comprehend and follow.
- The experimental setup and execution in the paper are adequate, resulting in promising outcomes and supporting the proposed approach.
- The visual results presented in the paper demonstrate good quality, further reinforcing the effectiveness of the proposed method.

**Weaknesses:**

1.

I observed a discrepancy between the equation presented in the paper and the implementation found in the provided demo code. This discrepancy, potentially caused by missing brackets and misrepresentation of the intended InfoCD, leads to a mismatch between the experimental results and the proposed idea. Consequently, concerns arise regarding the accuracy of the reported findings and the overall effectiveness of the proposed approach.

I have reviewed the provided code in 'loss_utils.py' and compared it to the equation mentioned in Section 3.2. I have identified a discrepancy in lines 197 and 198.

In the code, the calculation for l_infoCD(x_i, y_i) is implemented as "- torch.log(torch.exp(-0.2 * d1) + 1e-7 / torch.sum(torch.exp(-0.2 * d1) + 1e-7,dim=-1).unsqueeze(-1))". However, it appears that there are missing brackets in the expression. The correct calculation in Python should be "- torch.log((torch.exp(-0.2 * d1) + 1e-7) / torch.sum(torch.exp(-0.2 * d1) + 1e-7,dim=-1).unsqueeze(-1))" when the value of \tau is equal to 5.

Therefore, the issue lies in the missing brackets in the code implementation, which deviates from the equation provided in Section 3.2.

2.

Based on last question, I have concerns regarding the fairness of the comparison. Specifically, I would like to inquire whether both the baseline and the baseline + InfoCD models were trained and tested using identical settings, including training hyperparameters and the number of training epochs. My worry stems from the possibility that the observed improvement may be attributed to factors such as updates to the codebase, variations in training hyperparameters, or even longer training durations. This concern is amplified by the existence of a bug affecting the loss function in the provided code.

**Questions:**

See above

**Limitations:**

Limitations are discussed in the paper

---

> ### Author Rebuttal · Authors · 2023-08-07
>
> We sincerely thank the reviewer for valuable comments. Below we respond to reviewer concerns.
>
> **1. uploaded loss_utils.py code is incorrect:** Nice catch. Indeed this was not what was implemented. We maintained several versions and we apologize for accidentally uploading the incorrect version.
> **The implemented loss is exactly the regularized loss as in Eq. 6** used in all our experiments. \
> *The correct loss_utils.py* version has in lines 197 (as well as line 235), 198:
> >distances1 = 0.2 * d1 + 1e-7 * torch.log( torch.sum( torch.exp( -0.2 * d1 ), dim=-1 ).unsqueeze(-1) )\
> >distances2 = 0.2 * d2 + 1e-7 * torch.log( torch.sum( torch.exp( -0.2 * d2 ), dim=-1 ).unsqueeze(-1) ) \
> >where d1 and d2 denote two distance metrics in the original CD, and 1e-7 is a trade-off constant for the regularizer, which is universal across different datasets and networks.
>
> We emphasize that this is exactly Eq. 6 (modulo penalty parameter):
> $\mathcal{L_{\text{InfoCD}}}(x_i,y_i)=\frac{1}{\tau} \mathcal{L}_{\mbox{\small CD}}(x_i,y_i) + \lambda \mathcal{R}(x_i,y_i) $
>
> >where we have set $\lambda=1e-7$ in all our experiments. Notice that the first term on the right, $\frac{1}{\tau} \mathcal{L}_{\mbox{\small CD}}(x_i,y_i)$  corresponds to python code "0.2*d1" or equivalently "-torch.log(torch.exp(-0.2 * d1), dim=-1 ).unsqueeze(-1)" (This equivalent term bears similarity to the incorrect version and we believe is the source of our mistake.). The second term $\mathcal{R}(x_i,y_i)$ corresponds to python code "torch.log( torch.sum( torch.exp( -0.2 * d1 ), dim=-1 ).unsqueeze(-1) )".
>
>
> *Intuitive issue with uploaded version*:
> >Evidently, the misplaced brackets reviewer identified and our incorrectly uploaded version may not work. This is because "- torch.log(torch.exp(-0.2 * d1) + 1e-7 / torch.sum(torch.exp(-0.2 * d1) + 1e-7,dim=-1).unsqueeze(-1))" suppresses contribution from the additional term rendering the loss to behave similarly to vanilla CD loss (Note that the distances are normalized to the unit interval).
>
> We encourage the reviewer to verify our implementation by running the demo code by replacing the indicated lines.
>
> **2. Validation of our experimental results:** We ran our experiments with InfoCD based on the **default** hyperparameters such as learning rate and the maximum number of epochs in the public code, but only replacing the CD loss with our InfoCD loss. This has been clearly stated in the paper in L264-266: **"Hyperparameters such as learning rates, batch sizes and balance factors in the original losses for training baseline networks  are kept consistent with the baseline settings for fair comparisons."**.

---

> > ### Comment · Reviewer_mFMW · 2023-08-10
> > **Further concern about the equ 6 and the implementation**
> >
> > Thank you for the author's feedback. Based on the response, I have additional concerns regarding Equation 6 and its implementation.
> >
> > It appears that the paper does not provide any information about the value of lambda or an explanation for its usage. Equation 6 lacks a weighted regularization term. Moreover, the derivation of Equation 6 appears to be a result of simplifying Equation 5. The presence of lambda suggests a power operation of 1e-7 on the denominator, which requires further clarification. On the other hand, in point completion tasks, the typical magnitude of CD-L1 loss is around 1e-3, why add such a small weight to the regularization term?

---

> > > ### Author Response · Authors · 2023-08-11
> > > **Thanks for your concerns**
> > >
> > > **1. Eq. 6 lacks a weighted regularization term. Moreover, the derivation of Eq. 6 appears to be a result of simplifying Eq. 5:** Below are our responses:
> > >
> > > *(1) Decomposition in Eq. 6 as a Motivation for Scaling Regularization Term.* Eq. 6 shows that we can split our InfoCD loss into two components --- a CD loss plus a regularizer term. Drawing inspiration from this decomposition, we could re-weight the regularization term as is typical in practice.
> > >
> > > *(2) Conceptual Approach.* Nevertheless, we can conceptually ground the inclusion of penalty term and derive an expression with the penalty $\lambda$ by drawing direct inspiration from a line of recent works (see references [1, 2, 3] below for instance). These works propose an alternate variant of differential weighting between positive and negative pairs for the InfoNCE loss and show improved empirical results. Motivated by these works, let us modify our expression in Eq. 1. Namely, consider the following modified expression for Eq. 1, where we differentiate the pairs in the nominator and denominator by different temperature parameters $\tau', \tau$:
> > >
> > > $\mathcal{L}_{\text{InfoNCE}} = -\sum_x\log f(x, x^+, x^-; \tau', \tau)$ where
> > >
> > > $f(x, x^+, x^-; \tau', \tau) = \frac{\exp\left[-\frac{1}{\tau'}d(x^+,x;\theta)\right]}{\exp\left[-\frac{1}{\tau}d(x^+,x;\theta)\right]+\sum_{x^-}\exp\left[-\frac{1}{\tau}d(x^-,x;\theta)\right]}$, $\tau\geq\tau'>0$ and $d$ denotes a distance function. We can easily see that Proposition 1 for InfoNCE still holds for this modified formulation. Now, we can re-derive the expression in Eq. 5. Specifically,
> > >
> > > $
> > > \ell_{\text{InfoCD}}(x_i,y_i) = \frac{1}{\tau'|y_i|}\sum_k\min_jd(x_{ij},y_{ik}) + \log\left(\sum_k\exp\left[-\frac{1}{\tau}\min_jd(x_{ij},y_{ik})\right]\right) = \frac{1}{\tau|y_i|}\sum_k\min_jd(x_{ij},y_{ik}) + \lambda\log\left(\sum_k\exp\left[-\frac{1}{\tau}\min_jd(x_{ij},y_{ik})\right]\right),
> > > $ leading to
> > >
> > > $\mathcal{L_{\text{InfoCD}}}(x_i,y_i) = \frac{1}{\tau'} \mathcal{L_{\text{CD}}}(x_i,y_i) + \mathcal{R}(x_i,y_i) \propto \frac{1}{\tau} \mathcal{L}_{\text{CD}}(x_i,y_i) + \lambda \mathcal{R}(x_i,y_i),$ where $\lambda = \tau'/\tau$ is the ratio of the temperatures. Furthermore, the rest of the analysis (Lemma 1) in the paper follows in a straightforward way with this modified loss.
> > >
> > > *(3) References:*
> > >
> > > [1] Wang and Isola. "Understanding Contrastive Representation Learning through Alignment and Uniformity on the Hypersphere". In ICML, 2020.
> > >
> > > [2] Chuang et. al. "Debiased Contrastive Learning
> > > ". In NeurIPS, 2020.
> > >
> > > [3] Robinson et. al. "Contrastive Learning with Hard Negative Samples". In ICLR, 2021.
> > >
> > >
> > >
> > > **2. The presence of $\lambda$ suggests a power operation of 1e-7 on the denominator, which requires further clarification. On the other hand, in point completion tasks, the typical magnitude of CD-L1 loss is around 1e-3, why add such a small weight to the regularization term?** Please note that
> > >
> > > $$\log\left(\sum_k\exp\left[-\frac{1}{\tau}\min_jd(x_{ij},y_{ik})\right]\right) \approx \log|y_i|$$
> > >
> > > where $|y_i|$ denotes the number of points in the target point cloud $y_i$, and $|y_i|\approx1e4$ in our experiments for all the datasets. Therefore, we have $\log|y_i|\approx10$. Now, by substituting $\tau=5, \lambda=1e-7$ (as well as the typical magnitude of CD loss $1e-3$ as the reviewer suggested) into our InfoCD loss , we can easily calculate the magnitudes of the first and second terms in InfoCD are about $1e-4$ and $1e-6$, respectively, which is reasonable as typical regularization.
> > >
> > > In fact, we observe that the value of $\lambda$ is pretty robust within a large range. For instance, on the PCN dataset, using $\lambda=1e-3$ we can achieve 6.66 and 6.53 for InfoCD+PointAttN and InfoCD+SeedFormer, respectively, in contrast to 6.65 and 6.48 using $\lambda=1e-7$ in the paper. For simplicity, we set $\lambda=1e-7$ in all our experiments.

---

> > > > ### Author Response · Authors · 2023-08-18
> > > >
> > > > Dear Reviewer mFMW,
> > > >
> > > > Thanks for your valuable comments. We hope that our replies have well addressed your concerns about our submission. Please do let us know if you have more questions, and we will try to answer your questions asap.
> > > >
> > > > Again, we promise to release our code for reproducing our experimental results upon acceptance.

---

> > > > > ### Comment · Reviewer_mFMW · 2023-08-20
> > > > > **Thanks for the author's reply**
> > > > >
> > > > > Thank you for the author's patient responses. I believe my concerns have been mostly addressed, and I will update my rating to positive. I hope the author will consider adding more explanations to the supplementary material.

---

### Official Review · Reviewer_Lt2U · 2023-07-07

**Soundness:** 3 good
**Presentation:** 3 good
**Contribution:** 2 fair
**Rating:** 5
**Confidence:** 4

**Summary:**

The paper proposes a contrastive Chamfer distance to tackle the point cloud completion problem. The proposed CD loss maximizes the lower bound of the mutual information between two point cloud-based geometric surfaces, which leads to a more robust measurement of the similarities between two point clouds. On the other hand, the proposed CD loss is equivalent to adding a regularizer to the scaled CD, enabling a relaxed point alignment. Experiments of replacing CD with the proposed InfoCD in many state-of-the-art point completion models on MVP and some ShapeNet-based datasets show the good performance of the method.

**Strengths:**

- The introduction of the paper is concise and convincing. The authors have identified existing problems in the current research, and propose a solution based on these findings.

- The authors have conducted extensive experiments for the point cloud completion task on various datasets and have used the proposed loss function in different state-of-the-art methods.

- Although the analysis of the proposed CD loss is limited to point cloud completion tasks, the potential usage of the proposed loss function might be broader in various point cloud tasks.


**Weaknesses:**

- The authors claimed that the CD tends to have a hard constraint that points in the source point cloud should exactly lie on the points in the target point cloud. In contrast, InfoCD does not have this hard constraint. However, since usually the number of points in complete and partial point clouds is imbalanced, CD may not have this hard constraint. I wondered if a simple truncated CD would already solve this problem.

- In Line 179, “with another assumption that the matched point pairs keep unchanged over iterations”, which may not always be true. Any intuitions or experimental validations?

- Ablation study on $\tau$, lr is incomplete and confusing. The limitation is discussed but lacks some quantitative results for the efficiency analysis. The authors are encouraged to discuss the efficiency of the proposed method compared to the original CD loss.

- Table 1, 2, 3, 4 have inconsistent method comparisons. Could the authors provide more explanations?

- The application of the proposed CD loss is limited to point cloud completion. However, a broader discussion of other point cloud tasks could be discussed.


**Questions:**

Please see the detailed comments above. My main concerns are some unclear arguments and experiment settings.

**Limitations:**

The authors have mentioned some limitations in the conclusion section. However, a more detailed discussion is expected. Please see the detailed comments above.

---

> ### Author Rebuttal · Authors · 2023-08-06
>
> We sincerely thank the reviewer for the valuable comments. Below are our responses to the questions arising in the review:
>
> **1.1 A hard constraint on matching for CD and InfoCD:** We think that the reviewer may misunderstand this part. Firstly, we do not claim that InfoCD does not have such a constraint. In fact, due to the nature of nearest neighbor matching, InfoCD does have the hard matching constraint, i.e., one point in the source point cloud has a single match in the target point cloud. This is the same as CD. Secondly, both CD and InfoCD are applied to the reconstructed point cloud (NOT the input PARTIAL point cloud, as the review thought) and the complete point cloud, as distance metrics.
>
> **1.2 CD vs. Truncated CD vs. InfoCD:** Given the limited time, we implemented Truncated CD (T-CD) as T-CD = $\min(CD, thd)$ where $thd\in\{0.2, 0.4, 0.6, 0.8\}$ denotes a threshold, and tested T-CD on the ShapeNet-Part dataset used in the paper. This method achieves 4.72, 4.78, 4.88, and 4.75 in terms of L2-CD$\times10^3$, which can be slightly better than CD (4.82) but significantly worse than InfoCD (4.01).
>
> **2. Experimental evidence for the assumption in L179:** In the **newly uploaded PDF file** (please check the attachment), as a demonstration we plot some point correspondences during training over epochs (10,70,130), as shown in Fig. 1 where the blue points are ground truth and the red ones are predictions. InfoCD is able to help stabilize (i.e., keep unchanged) the (correct) correspondences much faster in training.
>
> **3. Training time and GPU memory footprint for computational efficiency:** InfoCD has only a few more operations than CD and thus in theory both computational efficiency should be similar. Numerically, for training CP-Net with CD and InfoCD per iteration it takes 0.4239$\pm$0.0019 and **0.4498$\pm$0.0030** second with 1052.627$\pm$0.0374 and **1053.692$\pm$0.0425** MB in GPU memory, respectively.
>
> **4. Inconsistancy in comparable methods in Table 1, 2, 3, 4:** We follow the literature, and aim to compare methods with public code as many as possible. Table 1 provides the results on PCN which is the most popular benchmark in the task of point cloud completion. Table 2 focuses on the diversity of models on MVP. For fair comparisons, we use the code public with the dataset to implement all the networks used for MVP. Tables 3 and 4 focus on Shapenet 55/34 that are recently proposed as benchmarks and smaller than the other datasets. We follow the previous works and choose a few representative networks to compare. On all the datasets with all the networks, our InfoCD consistently improves the performance.
>
> **5. A new point cloud task --- Single View Reconstruction (SVR):** Given the limited time, we add a new task of SVR that aims to reconstruct a point cloud from an image of the underlying object. Following 3DAttriFlow (*Wen et. al. "3D shape reconstruction from 2D images with disentangled attribute flow". In CVPR, 2022.*) and SnowflakeNet (*Xiang et. al. "Snowflake Point Deconvolution for Point Cloud Completion and Generation with Skip-Transformer". TPAMI, 2022.*), we sample 30k points from the watertight mesh in ShapeNet as the ground truth, and output 2048 points for evaluation based on per-point L1-CD$\times10^2$. We replace CD in SnowflakeNet with InfoCD for training, and list average comparison results below, demonstrating that InfoCD can generalize well for different tasks.
>
> -----------------------------
> Method                | Ave.
>
> -----------------------------
>
> 3DAttriFlow           | 3.02
>
> SnowflakeNet          | 2.86
>
> **InfoCD + SnowflakeNet | 2.73**
>
> -----------------------------

---

> > ### Author Response · Authors · 2023-08-18
> >
> > Dear Reviewer Lt2U,
> >
> > Thanks for your valuable comments. We hope that our replies have well addressed your concerns about our submission. Please do let us know if you have more questions, and we will try to answer your questions asap. Thanks

---

> > ### Comment · Reviewer_Lt2U · 2023-08-19
> > **Response to authors**
> >
> > Thanks for the authors' efforts in answering these questions.
> >
> > I have read all the comments by all reviewers and authors' responses. I think there remain some concerns in this work. I am optimistic that the work targets the fundamental problem in the loss function. However, the potential positive impact of point cloud-based research is questioned since the work still lacks solid evidence of generalizing to real-world applications. I appreciate that the authors have provided KITTI results that may prove---since only several numbers were provided---that InfoCD is effective in completing the car data in KITTI datasets. But it does not indicate the broader real-world applications of the proposed method. For example, the simplest task the authors could perform is a registration on KITTI as PCN was doing. Nevertheless, I think the paper could make it to the NeurIPS venue because of its effort in trying to improve the widely-used Chamfer loss, the experiments conducted on the point cloud completion task on various datasets, and the good performance it achieves on these datasets. However, I do not agree with reviewer eAft that the paper should be rated as "very strong accept" since the evaluation and the arguments still have some flaws.
> >
> > In response to the authors' response:
> >
> > 1. The "hard constraint" I referred to is mentioned in Figure 5 and related text. I understand that the proposed method still has this "constraint", but the authors claim the "constraint" to be less strict. In particular, the authors mentioned in Line 193-201 that CD forces the error between the ground truth and the predicted points to be zero during optimization while InfoCD only forces the error to be sufficiently small. However, in the real-world problem, the optimization of CD will not reach 0 error as expected yet yield good performance. The regularized CD as proposed in the paper converges to a lower value (as shown in Figure 5) but does not necessarily mean a better performance. I was thinking about whether the authors could rephrase these arguments and present a better way to explain the "weaker constraint" of the proposed CD. For example, a visualization of the real experiment data could be more convincing instead of showing illustrated Figure 5 (b) and (c).
> >
> > 2. The provided visualization does not show evidence that the correct correspondence will keep unchanged. I may be misunderstanding the figure. But I hope the authors could provide a better explanation.
> >
> > 3. The computational time and memory consumption seemed reasonable.
> >
> > 4. I have read the communication between the authors and the reviewer hcgP. I think the authors could further include a clearer explanation in the paper to clarify the experiment settings and the findings of the bug in the previous paper.
> >
> > 5. As I mentioned above, I appreciate the authors' efforts. But I do not think this will be sufficient evidence of broader applications.
> >
> > In conclusion, I would like to keep my rating.

---

### Author Rebuttal · Authors · 2023-08-08

We sincerely thank all the reviewers for their valuable comments. In summary,

1. We have responded to all the reviewer comments and uploaded a PDF file to show the point correspondences in training over epochs; this is based on the comment by Reviewer Lt2U.

2. We have added results from two new experiments: (1) a task of single-view reconstruction (SVR) for Reviewers Lt2U and eAft, and (2) KITTI results for Reviewers SCvJ and hcgP.

3. We have clarified a mistake in our demo code, as pointed out by Reviewer mFMW. **We will release our code for reproducing our experimental results upon acceptance.**

---

### Decision · Program_Chairs · 2023-09-21

**Decision:**

Accept (poster)

**Comment:**

The paper initially got mixed reviews. The authors submitted a solid rebuttal that helped in clarifying several issues. Unfortunately, the authors uploaded a wrong version of the source code, and caused confusion -- the reviewers accepted this as an unfortunate mistake and ignored the uploaded code. In the end, one champion for the paper and the rest of the reviewers supported this decision.

AC believes this could be a significant paper as it proposed a new metric that overcomes some of the shortcomings of the popular CD measure used in point cloud processing. Given the importance of the topic and the potential impact, the AC agrees with the reviewers and recommends this to be accepted.